# PiCCL: A lightweight multiview contrastive learning framework for image classification

**Yiming Kuang**[1], **Jianwu Guan**[2], **Hongyun Liu**[1,3], **Fei Chen**[1], **Zihua Wang**[4], **Weidong Wang**[1,3]*

**1** Research Center for Biomedical Engineering, Medical Innovation and Research Division, Chinese PLA General Hospital, Beijing, People's Republic of China, **2** Department of Radiology, First Medical Center, Chinese PLA General Hospital, Beijing, People's Republic of China, **3** Key Laboratory of Biomedical Engineering and Translational Medicine, Ministry of Industry and Information Technology, Beijing, People's Republic of China, **4** School of Biological and Medical Engineering, Beihang University, Beijing, People's Republic of China

☉ These authors contributed equally to this work.

* wangwd301@126.com

**citation:** Kuang Y, Guan J, Liu H, Chen F, Wang Z, Wang W. (2025) PiCCL: A lightweight multiview contrastive learning framework for image classification. PLoS One 20(8): e0329273 https://doi.org/10.1371/journal.pone.0329273

**Data availability statement:** The source code will be made publicly available after the acceptance of this manuscript. The code will be hosted in a public GitHub repository: https://github.com/YimingKuang/PiCCL.

## Abstract

We introduce PiCCL (Primary Component Contrastive Learning), a self-supervised contrastive learning framework that utilizes a multiplex Siamese network structure consisting of many identical branches rather than 2 to maximize learning efficiency. PiCCL is simple and light weight, it does not use asymmetric networks, intricate pretext tasks, hard to compute loss functions or multimodal data, which are common for multiview contrastive learning frameworks and could hinder performance, simplicity, generalizability and explainability. PiCCL obtains multiple positive samples by applying the same image augmentation paradigm to the same image numerous times, the network loss is calculated using a custom designed Loss function named PiCLoss (Primary Component Loss) to take advantage of PiCCL's unique structure while keeping it computationally lightweight. To demonstrate its strength, we benchmarked PiCCL against various state-of-the-art self-supervised algorithms on multiple datasets including CIFAR-10, CIFAR-100, and STL-10. PiCCL achieved top performance in most of our tests, with top-1 accuracy of 94%, 72%, and 97% for the 3 datasets respectively. But where PiCCL excels is in the small batch learning scenarios. When testing on STL-10 using a batch size of 8, PiCCL still achieved 93% accuracy, outperforming the competition by about 3 percentage points.

## 1 Introduction

In recent years, self-supervised learning (SSL) has gained significant popularity for its incredible performance while only using unlabeled data [1–6]. Since the noninvolvement of labels, SSL is a subcategory of unsupervised learning. However, today researchers commonly refer to SSL as its own category different from UL, with the distinction being SSL trains the network in a supervised manner using pseudo-labels, which are labels generated autonomously

**Funding:** This work was founded by the Scientific and Technological Innovation 2030 - "New Generation Artificial Intelligence" Major Project (2020AAA0105800). The funders had no role in study design, data collection and analysis, decision to publish, or preparation of the manuscript.

by pre-defined objectives [7,8]. It is common for networks trained by SSL to require a transfer learning step or fine-tuning step before applications like classification. In such cases, the role of the SSL is often referred to as pre-training or pretext task.

Contrastive learning (CL), a type of pretext task and a powerful tool for SSL, has been used in various fields including computer vision (CV) [1,2], natural language processing (NLP) [9,10], and graphs [5,11]. Most contrastive learning methods work by minimizing an objective function (loss function) which aims to bring the representations of positive sample pairs together and, often but not always, pushes representations of negative sample pairs apart. The latter's purpose is to prevent trivial solutions, but as SimSiam [12] demonstrated it's not always required. In the field of computer vision, positive and negative sample pairs refer to images with the same label and images with different labels respectively. For self-supervised approaches, due to the lack of real labels, the generation of sample pairs often relies on image augmentation, where views originating from the same sample image are positive pairs and views originating from different sample images are negative pairs, despite the source images might belong to the same category. The augmentations used might include color alternations, affine transformations, cropping and resizing, blurring and masking, etc., and are often applied randomly. So far, the majority of CL algorithms generate two views from each image per iteration, these two views form a positive pair, while all other views are considered to be negative samples. There are also algorithms that generates more than two views from each image, in this paper we categorize them as multiview contrastive learning algorithms (MCL). Since the number of views scales linearly with the number of views per image, the pairwise correlation calculation complexity scales quadratically. Thus a common problem for MCL is that their objective function is either complicated or asymmetric, making them hard to implement and compute. As most CL algorithms use other samples from the same batch as negative samples, most of them requires a large batch size to work well, making traning hardware intensive.

The search for better SSL algorithm is an ongoing task. Our research objective has been to develop methods that performs well, don't require large batch size, and can run on limited hardware. In this paper, we introduce **Pri**mary **C**omponent **C**ontrastive **L**earning (PiCCL), a new self-supervised contrastive learning framework for visual representation extraction. PiCCL is a MCL algorithm, it employs a symmetric multiplex Siamese network, which is an extension of the usual 2-fold structure to higher orders, it generates positive sample sets containing multiple views rather than positive pairs with just two views. To fully take advantage of this multiplex Siamese network without adding too much computation complexity, a brand-new loss function, Primary Component Loss (PiCLoss), is constructed. PiCLoss's idea is similar to other contrastive learning loss functions as it too works by promoting similarities within positive embedding sets (pair) while discouraging similarities between negative embedding sets (pair). The number of Siamese network branches, which is also the number of views per image, is "P", which can be set to any arbitrary integer greater than 1. To avoid the aforementioned scalability issue of MCL, PiCLoss first find the average (the primary component) of the embedding vectors of each set of positive samples, i.e. the set of views originated from the same image. And then calculate the instance discrimination network loss from the primary components. Thereby, retaining $\mathcal{O}(\mathcal{P})$ complexity. We tested PiCCL with $P = 4$ and $P = 8$ against popular algorithms, including previous state of the arts. PiCCL achieved highest accuracies in most of the tests, performing especially well in the small batch learning case.

To provide some intuition for PiCCL, consider the following. Suppose the learning has converged, and embeddings have formed into clusters in the embedding space. At the center of each cluster should be the defining features of that respective category. Let's call this cluster center the target embedding vector $\tilde{V}$. Now suppose the learning is in progress, the

embedding vector of sample $I$ is $f_{\theta_t}(I) = A$, here $f_{\theta_t}(*)$ represents the neural network and $\theta_t$ is the network weights at step $t$. Retrospectively we know we should update $\theta$ with the goal of bringing A toward $\tilde{V}$, but at this point $\tilde{V}$ is unknown, so the best we can do is to update $A$ toward the other positive embedding vectors, or, equivalently, update toward their average: $V$. The above is a stochastic process, the expected distance between the target embedding vector $\tilde{V}$ and the average embedding vector $V$ decreases as $P$ increases. i.e., when increasing $P$, the chance of the average embedding being closer to the true center is higher, so in each iteration, the updates to model weights are more effective.

Other than the above argument, PiCCL's multiplex Siamese structure also provides both more positive and negative samples per batch, making it more suited to small batch learning than more traditional SSL algorithms (Fig 1).

## 2 Related works

### 2.1 Contrastive learning

Most CV contrastive learning methods use Siamese networks or pseudo-Siamese networks. The former consists of twin networks of identical structure and weight, while the latter might have some subtle differences between the networks.

One of the most influential contrastive learning algorithms in recent years is SimCLR [13], it uses a twin Siamese network and achieved massive improvements compared to the State-Of-The-Art algorithms at that time. Siamese networks are a class of neural networks that take multiple (often 2) inputs, and process them by the same neural network (or equivalently, process them by neural networks with the same structure and shared weights). For a batch consisting of N images, SimCLR generates 2 views from each image via random image augmentation and forms an extended batch of size 2N, which is then fed into the Siamese network to obtain 2N embeddings. Then the network loss is calculated based on those embeddings using

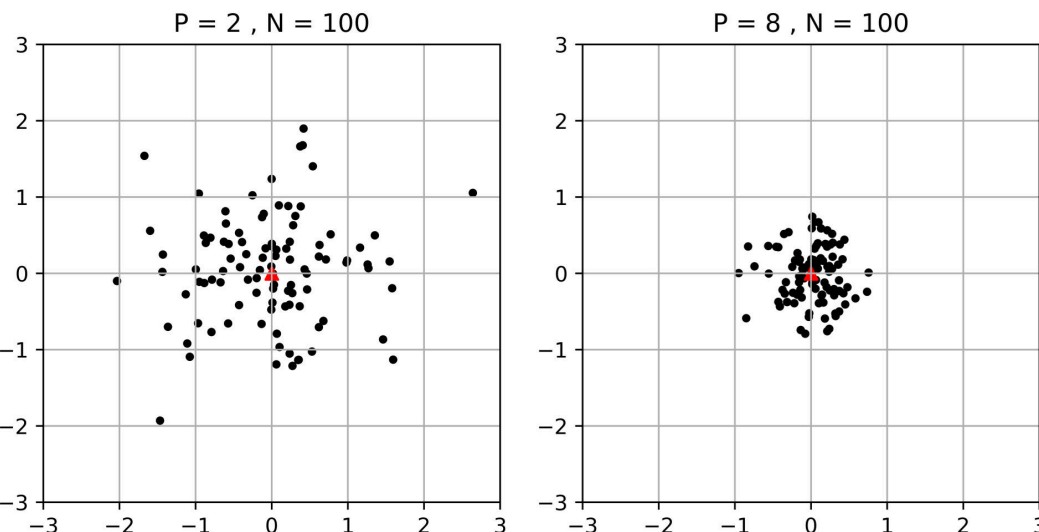

**Fig 1. Distribution of the average embeddings.** There are 100 black dots on each plot, each dot is the average position of P randomly generated points following a Gaussian distribution probability density ($\sigma = 1$, centered at (0,0)). For A, $P = 2$, and for B, $P = 8$. Comparing the 2 plots we can see, that as $P$ increases, the chance of the average representation being closer to the true center is higher, and thus the network updates are more efficient.

a loss function named NT-Xent (sometimes referred to as InfoNCE).

$$
\text{Loss} = \sum_{i=1}^{N} L_i = \sum_{i=1}^{N} \left( -log \frac{e^{(\vec{A}_i^1 \cdot \vec{A}_i^2)/\tau}}{\sum_{k=1}^{N} (e^{(\vec{A}_i^1 \cdot \vec{A}_k^2)/\tau} + e^{(\vec{A}_i^1 \cdot \vec{A}_k^1)/\tau}) - e^{(\vec{A}_i^1 \cdot \vec{A}_i^1)/\tau}} \right.
$$
$$
\left. -log \frac{e^{(\vec{A}_i^2 \cdot \vec{A}_i^1)/\tau}}{\sum_{k=1}^{N} (e^{(\vec{A}_i^2 \cdot \vec{A}_k^1)/\tau} + e^{(\vec{A}_i^2 \cdot \vec{A}_k^2)/\tau}) - e^{(\vec{A}_i^2 \cdot \vec{A}_i^2)/\tau}} \right) \tag{1}
$$

Eq (1) is the NT-Xent loss function for a batch of images, its form has been reformulated to keep the notation consistent within this literature. $\vec{A}_i^1$ is the L2 normalized embedding vector of the first view of the $i$-th image, and similarly $\vec{A}_k^2$ is the L2 normalized embedding vector of the second view of the $k$-th image, $N$ is the number of images within a batch, i.e. batch size. These notation will be used through out this paper. $\tau$ is the "temperature" parameter. The numerator term of NT-Xent is the "attractive terms" that brings positive sample pairs together, while the denominator terms are the "repulsive terms" that pushes negative pairs apart.

Other algorithms like Barlow-Twins [14], and VICReg [15], also use the same kind of symmetric Siamese network but differentiate themselves through their unique loss functions.

$$
\text{Loss} = \sum_{i=1}^{N} \left( 1 - \vec{A}_i^1 \cdot \vec{A}_i^2 \right)^2 + \alpha \sum_{i=1}^{N} \sum_{\substack{j=1 \\ j \neq i}}^{N} \left( \vec{A}_i^1 \cdot \vec{A}_j^2 \right)^2 \tag{2}
$$

Eq (2) is Barlow-Twins' loss function, named $L_{BT}$ [14]. Its first term is the "attractive term" while the second term is the "repulsive term", $\alpha$ is a constant parameter controlling the strength of the repulsive term.

The aforementioned methods require positive pairs for feature learning, as well as negative pairs to prevent collapsing. On the contrary, methods like SimSiam [12], SwAV [16], and BYOL [17] use only positive pairs and thus are capable of online learning. To prevent collapsing, all the above features some asymmetries between the two branches, SimSiam uses a symmetric Siamese network just like SimCLR, but features a predictor head on one branch while employing a stop gradient operation on the other. SwAV and BYOL feature twin networks consisting of the same structure but different weights, such networks are sometimes referred to as pseudo-Siamese networks.

$$
\text{Loss} = \frac{1}{N} \sum_{i=1}^{N} \left( \frac{1}{2} \left( D(\vec{A}_i^1) \cdot P(\vec{A}_i^2) \right) + \frac{1}{2} (D(\vec{A}_i^2) \cdot P(\vec{A}_i^1)) \right) \tag{3}
$$

Eq (3) is the Loss function of SimSiam, $D(*)$ is the stop gradient (or detach) operation which removes the argument from the backpropagation computation graph. $P(*)$ is the predictor operation which in SimSiam's original implementation is a 2 layer MLP.

More efficient pretext tasks leads to better training outcomes. Unrepresentative positive sample pairs, such as sample pairs that don't contain the same object due to random crop, leads to inefficient training. SemanticCrop [18] and ScoreCL [19] are extension methods that can be used alongside popular methods like SimCLR and Barlow-Twins. Semantic-Crop introduced a new weighted crop method that replaces the common random crop image augmentation method. It uses center-suppressed probabilistic sampling to favor crops that are

dissimilar yet still lands in the target area. ScoreCL addresses this problem by adding an additional term to the loss function that weights the network loss based on how dissimilar the positive pairs are. PiCCL attempts to address this problem by increasing the number of views, the more views a positive sample set contains, the less likely it is to be unrepresentative.

## 2.2 Multi-view contrastive learning

There have been works trying to expand the 2-fold Siamese networks used by traditional contrastive learning algorithms to higher orders, some examples include CMC [20], K-shot [21], LOOC [22], and E-SSL [23]. In this paper, we refer to this family of algorithms as multiview contrastive learning (MCL) algorithms, and PiCCL is one of them. K-Shot's framework is closest to PiCCL, it features multiple identical branches just like ours. However, its loss function is fairly complicated and computationally heavy, requiring eigenvalue decomposition calculations for each and every view, this step itself is $\mathcal{O}(P^3)$, which significantly adds to the training cost. LOOC's pretext tasks are asymmetric, each branch uses one type of predetermined image augmentation method. Its loss function is a generalization of InfoNCE, it calculates the InfoNCE loss on every possible pair of branches and finds their average, due to the P choose 2 combination, this loss function is $\mathcal{O}(P^2)$. E-SSL is an extension method that can be used on other self-supervised learning methods, it creates 4 additional branches on top of the SSL Siamese framework, each branch uses a predetermined angle of rotation as its pretext task. For the added branches, predictions are made on the angle of rotation, the error is calculated and used for backpropagation. Contrastive multiview coding (CMC) is one of the earlier methods in multiview contrastive learning, it requires the use of multimodal data. Each branch is tasked to extract feature from one mode, and the network aims to learn common features across the different modes. In the original work, multimodal views of ordinary RGB pictures are obtained by converting it to Lab format, and then treat the L channel and the ab channel as different modes. The paper provides 2 loss functions for networks with $P > 2$, one have a "core" view" where other views are compared against, another have no "core" view and comparisons are made for every possible pairs. The former one is $\mathcal{O}(P)$ and the latter $\mathcal{O}(P^2)$.

The above methods all suffer more or less from at least one of the following problems: (1) complicated network structure, this includes the use of asymmetric pretext tasks or network weights; (2) computationally intensive, especially when P is large; (3) requires multimodal data, which could hurt generalizability. Motivated by the above, we designed PiCCL to be a simple, scalable, and generalizable algorithm (Table 1).

## 2.3 Small batch learning

Online learning is a sub-field of machine learning where data arrives in a sequential order, and the neural network updates its parameters with the objective of making a better prediction on the next sample. This is in contrast with traditional machine learning approach which the entire dataset is available from the start. Small batch learning, on the other hand, refers to mini-batch learning with a small batch size. Contrary to the name, mini-batch sometimes aren't mini at all, for example some tests in [14] uses a batch size of 4096. Both online learning and small batch learning have a substantial advantage in computational cost, most significantly, memory requirement, as loading a small batch of samples requires very little memory. Also, small batch size makes data acquisition possible on a single device as gathering a large batch of data, especially if we wish it to contain a wide range of different classes, is both hard and time consuming. These advantages make online learning and small batch learning ideal for off-line mobile device learning, low-power training and high dynamic environment learning.

**Table 1. Summary of related works.**

| Name | number of branches (P) | symmetric | multimodal data | complexity(P) |
|---|---|---|---|---|
| SimCLR [13] | 2 | yes | no | - |
| Barlow-Twins [14] | 2 | yes | no | - |
| VICREG [15] | 2 | yes | no | - |
| SimSIam [12] | 2 | no | no | - |
| SwAV [16] | 2 | no | no | - |
| BYOL [17] | 2 | no | no | - |
| CMC [20] | any | no | yes | $\mathcal{O}(P) \, / \, \mathcal{O}(P^2)$ |
| K-shot [21] | any | yes | no | $\mathcal{O}(P^3)$ |
| LOOC [22] | any | no | no | $\mathcal{O}(P^2)$ |
| E-SSL [23] | 6 | no | no | - |
| PiCCL (Ours) | any | yes | no | $\mathcal{O}(P)$ |

Symmetric: whether the network or the loss function contains branch specific terms that will cause the error to propagate differently between branches. For example, consider the stop gradient operator in SimSiam's loss function (3). Complexity(P): loss function's time complexity with respect to the number of network branches, this is invalid for algorithms with fixed branch number.

Generally speaking, aside from methods like SimSiam which doesn't utilize negative pairs, most SSL methods require a large batch size to work well, as it provides more negative samples. Most SSL algorithms use batch sizes larger than 128, which could be difficult for offline on-device incremental learning tasks.

# 3 Method

The forward pass flow diagram of PiCCL is illustrated in Fig 2. PiCCL contains P identical branches, each containing the same encoding and decoding network. The image augmentation module takes a batch of N images and outputs an extended batch composed of P augmented batches each containing N views. The views are first mapped onto the latent space by the encoder and then mapped onto the embedding space by the decoder. Finally, network loss (error) is calculated from the L2 normalized embedding vectors using a custom loss function: PiCLoss. This concludes the forward pass and backpropagation can be initiated.

## 3.1 Image augmentation

The choice of image augmentation methods is crucial for the quality of representation learning [24]. A good augmentation should retain as much relevant information as possible while altering the rest. Here, we implement a slightly modified version of the image augmentation process used by Barlow-Twins, which consists of random applications of (1) cropping followed by resizing to the original size, (2) horizontal flip, (3) color jitter, (4) transform to grayscale, and (5) Gaussian blur. The same image augmentation paradigm is applied to all Siamese network branches. The list of augmentations is listed in Table 2.

## 3.2 Loss function

PiCLoss, the loss function designed for PiCCL, is expressed as Eq (4b).

$$\vec{V}_n = \frac{\frac{1}{P} \sum_p \vec{A}_n^p}{||\frac{1}{P} \sum_p \vec{A}_n^p||_2} \tag{4a}$$

$$\text{Loss} = \left\langle 1 - \vec{A}_n^p \cdot \vec{A}_n^q \right\rangle + \alpha \left\langle e^{|\vec{V}_n \cdot \vec{V}_m|} \right\rangle \tag{4b}$$

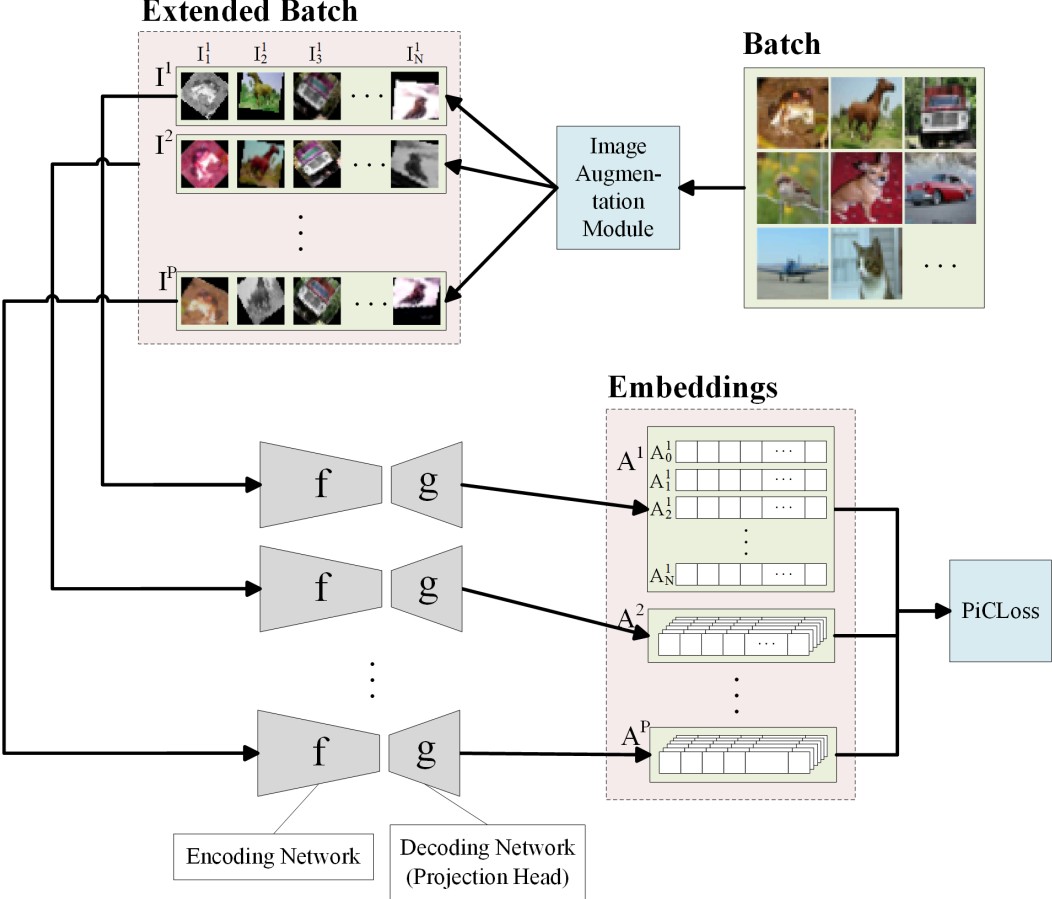

**Fig 2. Network Structure of Primary Component Contrastive learning (PiCCL).** PiCCL is a multiview contrastive learning algorithm, *P* views are generated from each sample image and fed into a multiplex Siamese network. The latent space embedding vectors are obtained and network loss is calculated with a custom function: PiCLoss.

**Table 2. Image augmentation methods.**

| Methods | source | params | description |
|---|---|---|---|
| random crop and resize | torchvision | default | crop the image then resize to original size |
| random horizontal flip | torchvision | p = 0.5 | flip the image horizontally with probability p |
| random apply of color jitter | Barlow-Twins | p = 0.8 brightness = 0.6, contrast = 0.6, saturation = 0.6 hue = 0.2 | make changes to brightness, contrast, saturation and hue with probability p |
| random grayscale | torchvision | 0.5 | turn image into grayscale with probablity p |
| random gaussian blur | Barlow-Twins | P = 1 | apply gaussian blur with probability p |

The list of image augmentation methods used by PiCCL. For source, "torchvision" means the function is from torchvision 0.20, "Barlow-Twins" means the function os from Barlow-Twins' published source code.

Same as before, $\vec{A}_n^p$ denotes the L2 normalized embedding vector of the p-th view of the n-th image. The superscript *p* is no longer confined to 1 or 2, but can now be any natural number from 1 to *P*. $\alpha$ is the regularization parameter, the idea is that PiCLoss can be

tweaked to fit different scenarios. However, during testing, we found that setting $\alpha = 2$ accommodates most cases. $\langle * \rangle$ is the average value operation. $\vec{V}_n$ are the "Primary Components" in PiCCL's name, it is the average of embeddings of all views from the same image followed by a $L_2$ normalization.

The first term in PiCLoss is the attractive term. $S_n = \vec{A}_n^p \cdot \vec{A}_n^q$ is a P-by-P symmetric matrix that represents all pair-wise cosine similarity between embedding vectors of views originating from image $n$. Cosine similarity are calculated as the dot product of the L2 normalized embedding vectors, values are bounded by –1 and 1, and diagonal elements are always 1.

The second term in PiCLoss is the repulsive terms that penalizes similarities between primary components of different images, quantified by $|\vec{V}_n \cdot \vec{V}_m|$. By calculating the primary components first, this $\vec{V}_n \cdot \vec{V}_m$ matrix only contains N-by-N elements. An absolute value operation is applied since negative correlations should also be penalized, and an exponentiation operation is applied to help with the uniformity of $\vec{V}_n \cdot \vec{V}_m$ as it addresses a much more substantial cost on extreme values (Fig 3).

## 4 Results

To test PiCCL's performance, we benchmarked it against popular algorithms including Sim-CLR, Barlow-Twins, SimSiam, and E-SSL. We used 2 flavors of PiCCL: PiCCL(4) with 4 network branches and PiCCL(8) with 8 branches. We attempt to make the comparisons fair and

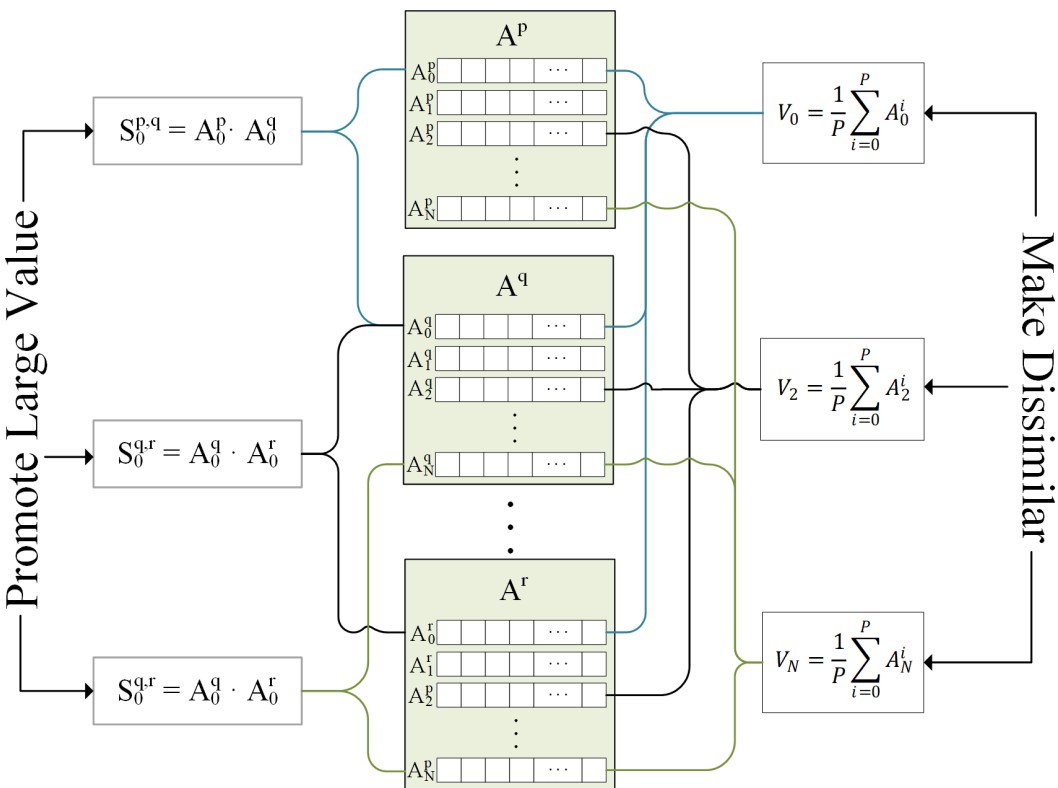

**Fig 3. Visualization of PiCLoss.** Two sets of operations are applied to the L2 normalized embedding vectors (middle boxes). The left side is the attractive term promoting cluster formation. The right side is the repulsive term decorrelating negative pair embeddings.

representative, that means using the same networks, augmentations, and hyperparameters whenever possible. The encoder of choise is ResNet-18 [25], the smallest variant of the ResNet family and a very popullar choice in the computer vision community. To accommodate the picture size of CIFAR-10, CIFAR-100, and STL-10, we have made the following changes to the ResNet18 encoder. For CIFAR-10 and CIFAR-100 images, whose size is 32 by 32, the first 7∗7 convolution layer is replaced with a 3∗3 stride = 1 layer. The first max pooling as well as the final fully connected layer is removed. These changes are the same as SimCLR's implementation on CIFAR-10. For STL-10 images, whose size is 96 by 96, the first 7∗7 convolution layer is replaced with a 5∗5 stride = 2 layer, and the final fully connected layer is removed.

For training of PiCCL, SimCLR, and Barlow-Twins, a 2-layer MLP projector is added to the encoder (first layer: 512 to 2048, second layer 2048 to 128). For training of SimSiam, we directly followed its literature, using the same 3-layer MLP as the projector and the 2-layer MLP as the predictor.

For all tests, neural network are trained for 1000 epochs, learning rate are set to 0.6∗batchsize/64 unless specified. Learning rate decreases following a cosine annealing schedule. For the loss functions, we set the hyper parameters according to the original literatures. For $NT-Xent$ 1, the temperature parameter is set to 0.5. For $L_{BT}$, $\lambda$ is set to 0.5. For CIFAR-10 and CIFAR-100, we also conducted test for E-SSL. The model is ResNet18, with the same changes applied. Other parameters are kept as the same as its original literature.

Accuracies are evaluated using the linear classifier method. The projector (and predictor for SimSiam) are discarded, and a linear layer is attached after the encoder CNN for classification. For each test, the classifier layer is trained for 100 epochs using the training set. After each epoch, the accuracy is evaluated using the testing set, and the highest accuracy across the 100 epochs is reported. The batch size is set to 256, the and learning rate is set to 0.6, no learning rate scheduling is used.

## 4.1 STL-10

STL-10 contains 3 sets of images of size 96 by 96: a "unlabeled" set containing 100,000 unlabeled images of various catagories; a "train" set containing 5,000 labeled images, 10 classes, 500 images each; and a "test" set containing 8,000 labeled images, 800 for each class. For our test, we used the "unlabeled" set to train the network, after which the projector (and predictor) is discarded, then we used the "train" set to train the linear classifier, and finally, use the "test" set to evaluate accuracy.

Table 3 displayed the linear classifier accuracies of SimCLR, Barlow-Twins, SimSiam, PiCCL(P = 4), and PiCCL(P = 8). The batch size is set to 256, a commonly accepted size for SSL algorithms of a similar kind. At all epochs, both PiCCL(4) and PiCCL(8) outperformed other methods. Among the 2 variants of PiCCL, We did not observe any overall trend to whom is superior. The highest overall accuracy is achieved by PiCCL(4) with a score of

**Table 3. Results on STL-10 with batch size = 256.**

| Methods | 100epoch | 200epoch | 500epoch | 1000epoch | Best |
|---|---|---|---|---|---|
| SimCLR | 95.17 | 96.10 | 96.75 | 96.79 | 96.79 |
| Barlow-Twins | 93.14 | 94.33 | 95.86 | 95.97 | 95.97 |
| SimSiam | 84.81 | 88.95 | 87.07 | 86.05 | 88.95 |
| PiCCL (4) | 95.25 | **96.50** | 96.83 | **97.55** | **97.55** |
| PiCCL (8) | **95.79** | 96.33 | **97.05** | 97.18 | 97.18 |

The one trial accuracy of various methods on the STL-10 dataset while batch size is held at 256. The highest accuracy of each column is highlighted in bold, the values are percentage points.

97.55%. Model accuracy had mostly stabilized by 500 training epochs, therefore we selected this duration as the optimal training length for subsequent experiments.

Table 4 displayed the accuracies of the algorithms at 500 epochs, with batch size set to 4, 8, 16, 64, and 256. This table is meant to showcase the effect of batch size on classification accuracies. At the extreme case of N = 4, Barlow-Twins took the first place in accuracy, SimSiam took the second place followed by PiCCL(8). For batch sizes of 8 and onward, both PiCCL(4) and PiCCL(8) outperforms all other methods.

To strengthen our findings, we repeated the N = 8 and N = 256 tests 5 times and analyzed the significance using the independent samples T test. The results are displayed in Fig 4.

For N = 8, both PiCCL(4) and PiCCL(8) showed a statistically highly significant ($p \leq 0.001$) advantage over SimCLR, Barlow-Twins and SimSiam. There is also a significant difference ($p \leq 0.01$) between PiCCL(4) and PiCCL(8). PiCCL(8) outperformed SimCLR

**Table 4. Results on STL-10 at 500 epoch.**

| Methods | N = 4 | N = 8 | N = 16 | N = 64 | N = 256 |
|---|---|---|---|---|---|
| SimCLR | 84.44 | 88.98 | 95.15 | 96.71 | 96.75 |
| Barlow-Twins | **89.48** | 90.35 | 91.20 | 91.90 | 95.86 |
| SimSiam@200[1] | 88.95 | 88.95 | 88.95 | 88.95 | 88.95 |
| PiCCL (4) | 88.00 | 92.91 | 95.39 | 96.81 | 96.83 |
| PiCCL (8) | 88.66 | **93.54** | **95.67** | **97.02** | **97.05** |

The one trial accuracy of various methods on the STL-10 dataset with varying batch sizes, training lasted 500 epochs. The highest accuracy of each column is highlighted in bold, the values are percentage points.
[1] We did not test SimSiam with varying batch sizes as SimSiam's loss function is unaffected by batch size. The reported value is the highest value we obtained from previous test.

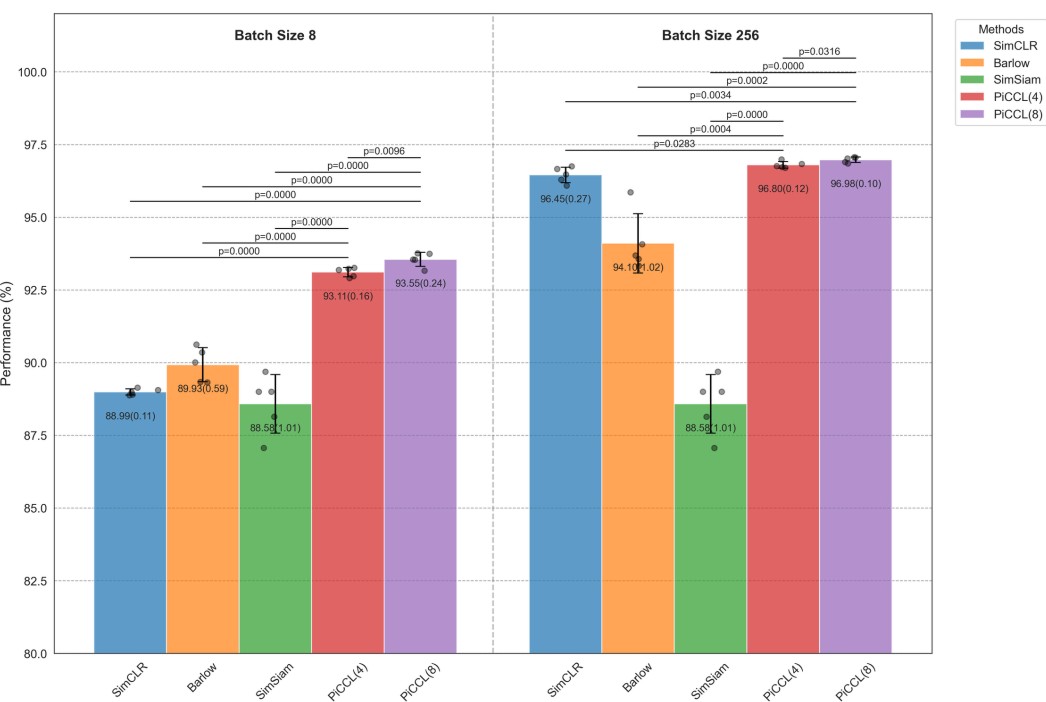

**Fig 4. Results on STL-10 at 500 epoch with batch size = 2 and 256.** The left panel compares the methods with batch size = 8, the right panel compares the methods with batch size = 256. each test is repeated 5 times, the mean(std) is written on the columns, the p-values are drown on the displayed on the top of the figure.

by 4.56% (95%CI:[4.28, 4.83]), outperformed Barlow-Twins by 3.62% (95%CI:[2.96, 4.28]), outperformed SimSiam by 4.97% (95%CI:[3.89, 6.03]).

For N = 256, PiCCL(4) and PiCCL(8) significantly outperformed SimCLR at P values less than 0.05 and 0.01 respectively. The advantage over Barlow-Twins and SimSiam is statistically highly significant ($p \leq 0.001$). Between the two variants, PiCCL(8) slightly outperforms PiCCL(4) with a P-value less than 0.05.

Shrinking the batch size from 256 to 8 caused performance degradation to all methods except SimSiam. PiCCL(8) degraded by 3.43% (95%CI:[3.16, 3.70]) and PiCCL(4) degraded by 3.69% (95%CI:[3.48, 3.90]). While SimCLR degraded by 7.46% (95%CI:[7.16, 7.76]) and Barlow-Twins degraded by 4.17% (95%CI:[2.95, 5.39]).

## 4.2 CIFAR-10 & CIFAR-100

The CIFAR-10 and CIFAR-100 datasets are very simillar, they both consists a training set of 50,000 and a testing set of 10,000 labeled RGB images, each of size 32 by 32. Their difference being CIFAR-10 contains 10 classes while CIFAR-100 contains 100 classes. We use the "train" set for both unsupervised training of the network, and supervised finetuning of the linear classifier, we use the "test" set for accuracy evaluation.

For the tests on CIFAR-10 and CIFAR-100 (Table 5), both PiCCL(4) and PiCCL(8) consistently outperformed the other algorithms. When testing under the same settings, SimSiam suffered from severe over fitting, this is likely due to CIFAR datasets having both less samples and less pixels per sample. To make a fair comparison we decreased SimSiam's learning rate, its accuracy at 500 epoch is lower than 200 epoch, indicating that albeit to a lesser degree, the problem of over fitting is still present.

## 5 Discussion

### 5.1 Performance of PiCCL

PiCCL performs great in the general setting. When testing on STL-10 with batch size of 256 (Table 3), PiCCL achieved 97.55% accuracy, outperforming all other methods tested, including previous state of the arts. We think the primary contributor to the performance gain is the combination of the multiplex Siamese network structure and our unique loss function: PiCLoss. Each positive sample set contains $P$ positive samples, rather than 2 in other methods, which provides a stronger and more confident anchor for the positive views. We also think the exponential regularization played a crucial role since in our own testing we find that changing it to $L_1$ or $L_2$ regularization resulted in performance degradation.

**Table 5. Results on CIFAR-10 & CIFAR-100.**

| | CIFAR-10 | | CIFAR-100 | |
|---|---|---|---|---|
| Methods | 200 Epoch | 500 Epoch | 200 Epoch | 500 Epoch |
| SimCLR | 92.25 | 93.26 | 69.71 | 71.71 |
| Barlow-Twins | 90.74 | 92.52 | 66.66 | 68.22 |
| SimSiam[1] | 82.92 | 79.85 | 54.20 | 50.93 |
| E-SSL | 90.37 | 93.00 | 64.23 | 69.09 |
| PiCCL (4) | 93.15 | 93.61 | 72.10 | **72.75** |
| PiCCL (8) | **93.73** | **94.04** | **72.38** | 72.44 |

The highest accuracy of each column is highlighted in bold, the values are percentage points.
[1]The numbers presented here for SimSiam has the learning rate decreased by a factor of 4.

PiCCL's performance is robust to small batch sizes (Table 4). PiCCL's performance is sub optimal for N = 4, but at N = 8, PiCCL(4) and PiCCL(8) already reached 92.91% and 93.54%. Meanwhile, SimCLR only obtained 88.98%, roughly 4 percentage points lower than PiCCL. We think having more views when calculating network loss is a key factor to this performance edge. First, more positive samples provide a more accurate and representative target embedding vector (the primary component) for which views are updated towards. And second, more negative samples mean the views have more instances to discriminate against, which also improves training quality. This theory is in line with our observation that PiCCL(8) outperforms PiCCL(4) in small batch settings.

The number of network branches (P) can take any number greater or equal to 2. In our tests, PiCCL(2) uses the least amount of system resources, but its performance lags behind PiCCL(4), further suggesting that PiCCL's performance increases with P. PiCCL(16) on the other hand, uses more system resources than PiCCL(8) while producing on par performances, suggesting a limit to this scaling. On the dataset and the neural network we used, PiCCL(4) and PiCCL(8) represent the "sweet spot" that balances performance and computation complexity. On other datasets and neural networks, this observation may vary.

## 5.2 Efficiency of PiCCL

To validate our claim that PiCCL is relatively lightweight and scalable, we recorded the training time of the first 5 epochs on STL-10, deduced the average time per epoch per network branch, and reported the results in Table 6. PiCCL(2P) takes roughly twice as much time as PiCCL(P). PiCCL(2)'s training speed is on par with SimCLR and Barlow-Twins while PiCCL(4) is on par with SimSiam. The training time per network branch remains fairly constant across ALL PiCCL variants, validating our claim that PiCCL scales linearly with P. Table 6 also displays the video card memory usage. PiCCL(2) uses the least amount of memory among all methods. PiCCL(4) uses roughly 60% more than PiCCL(2), which is still less than SimSiam. The memory per branch slightly decreases as P increases, suggesting that PiCCL's memory usage also scales linearly with P.

## 5.3 Weakness of our study

Due to both computer hardware constraints and time constraints, we weren't able to test PiCCL on more complex and comprehensive datasets like ImageNet. We weren't able to test PiCCL with larger batch size or a larger P. We also have to use ResNet-18 as our encoder backbone rather than the more common ResNet-50. However we think the above tests are enough to showcase the efficacy of our algorithm.

**Table 6. Speed and Memory Metrics.**

|  | Time per epoch | Time per epoch per branch | memory | memory per branch |
|---|---|---|---|---|
| SimCLR | 43.2 | 21.6 | 1463 | 731.5 |
| B-Twins | 43.2 | 21.6 | 1467 | 733.5 |
| SimSiam | 80.2 | 40.1 | 2719 | 1359.5 |
| PiCCL(2) | 43.2 | 21.6 | 1317 | 658.5 |
| PiCCL(4) | 83.4 | 20.85 | 2091 | 522.75 |
| PiCCL(8) | 162.4 | 20.3 | 3783 | 472.88 |
| PiCCL(16) | 325.8 | 20.36 | 6855 | 428.44 |

The time reported has units in seconds, and the memory reported has units in megabytes (MiB). The dataset tested is STL-10 and the batch size is 64.

### 5.4 PiCCL's use case and future work

PiCCL's great performance under ordinary circumstances (Table 3) makes it a viable solution for computer vision self-supervised learning tasks. Our near future objective is to further optimize PiCCL by integrating more powerful and more recent encoders such as vision transformers [2], experiment on different loss function regularization, and bring PiCCL to other data types like audio. We will also attempt to solve real world problems using PiCCL as deep learning has been proven to be affective in countless cases [26–32]. In the long run, we want to incorporate PiCCL into scenarios where its small batch learning capabilities can be utilized. Referring to Table 4, When P = 8, PiCCL outscored the competition by more than 3 percentage points, which makes PiCCL especially suited for online styled small batch learning tasks. For example, low power device learning where memory is a constraint, or offline learning where gathering a large batch of distinct samples is hard.

## 6 Conclusion

In this paper, we proposed a simple and lightweight multiview contrastive learning algorithm called PiCCL. PiCCL uses a multiplex neural network structure, the pretext tasks and network weights are shared between network branches, making it a true Siamese network. PiCCL's unique loss function "PiCLoss" simplifies computation by using primary components as a middle step, and thus can retain $\mathcal{O}(P)$ complexity. We benchmarked PiCCL on STL-10, CIFAR-10, and CIFAR-100. In our tests, PiCCL outperformed SimCLR, Barlow-Twins, and SimSiam most of the time, especially in small batch size settings. Our vision for PiCCL involves applying it on image classification tasks where batch size is constrained, like on-device-learning or online-learning scenarios, as well as using PiCCL with larger state of the art models, these will be our future focus.

## Author contributions

**Conceptualization:** Yiming Kuang, Jianwu Guan, Hongyun Liu, Fei Chen, Zihua Wang, Weidong Wang.

**Funding acquisition:** Weidong Wang.

**Methodology:** Yiming Kuang.

**Software:** Yiming Kuang.

**Supervision:** Weidong Wang.

**Writing – original draft:** Yiming Kuang.

**Writing – review & editing:** Yiming Kuang, Weidong Wang.

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
