## [Decision Letter · Decision Letter 0]

3 Dec 2024

PONE-D-24-42971PiCCL: a lightweight multiview contrastive learning framework for image classificationPLOS ONE

Dear Dr. Kuang,

Thank you for submitting your manuscript to PLOS ONE. After careful consideration, we feel that it has merit but does not fully meet PLOS ONE’s publication criteria as it currently stands. Therefore, we invite you to submit a revised version of the manuscript that addresses the points raised during the review process.

We look forward to receiving your revised manuscript.

Kind regards,

Tianlin Zhang

Academic Editor

PLOS ONE

Journal requirements: When submitting your revision, we need you to address these additional requirements. 1. Please ensure that your manuscript meets PLOS ONE's style requirements, including those for file naming. The PLOS ONE style templates can be found at https://journals.plos.org/plosone/s/file?id=wjVg/PLOSOne_formatting_sample_main_body.pdf and https://journals.plos.org/plosone/s/file?id=ba62/PLOSOne_formatting_sample_title_authors_affiliations.pdf 2. Please note that PLOS ONE has specific guidelines on code sharing for submissions in which author-generated code underpins the findings in the manuscript. In these cases, all author-generated code must be made available without restrictions upon publication of the work. Please review our guidelines at https://journals.plos.org/plosone/s/materials-and-software-sharing#loc-sharing-code and ensure that your code is shared in a way that follows best practice and facilitates reproducibility and reuse. 3. We note that the grant information you provided in the ‘Funding Information’ and ‘Financial Disclosure’ sections do not match.  When you resubmit, please ensure that you provide the correct grant numbers for the awards you received for your study in the ‘Funding Information’ section. 4. Thank you for stating the following financial disclosure:  [This work was founded by the Scientific and Technological Innovation 2030 - ”New Generation Artificial Intelligence” Major Project (2020AAA0105800).].  Please state what role the funders took in the study.  If the funders had no role, please state: ""The funders had no role in study design, data collection and analysis, decision to publish, or preparation of the manuscript."" If this statement is not correct you must amend it as needed. Please include this amended Role of Funder statement in your cover letter; we will change the online submission form on your behalf. 5. Please upload a copy of S1 Appendix, S2 Appendix and S3 Appendix. to which you refer in your text on page 22. Please amend the file type to 'Supporting Information'. If the Supplementary file is no longer to be included as part of the submission please remove all reference to it within the text.

Reviewers' comments:

Reviewer's Responses to Questions

**Comments to the Author**

1. Is the manuscript technically sound, and do the data support the conclusions?

Reviewer #1: Partly

Reviewer #2: Yes

2. Has the statistical analysis been performed appropriately and rigorously? 

Reviewer #1: N/A

Reviewer #2: Yes

3. Have the authors made all data underlying the findings in their manuscript fully available?

Reviewer #1: No

Reviewer #2: Yes

4. Is the manuscript presented in an intelligible fashion and written in standard English?

Reviewer #1: Yes

Reviewer #2: Yes

5. Review Comments to the Author

Reviewer #1: • The introduction begins well but concludes abruptly. To improve consider summarizing the main contributions of the work at the end of the introduction. This will provide a good transition into the subsequent sections.

• Some technical terms such as "pseudo-labels" and "pretext tasks" may not be familiar to all readers. Providing brief definitions or explanations for these terms will make the paper accessible to a broader audience.

• The explanations for Figure 1 and Figure 2 currently in the introduction can be relocated to Section 3 (Methodology) for better alignment with the context. This will also enhance the logical flow of the manuscript.

• Ensure that all figures are explicitly referred to within the text for clarity and better reader navigation.

• In S2 Appendix, the manuscript briefly describes hyperparameters used for training. Explain how these values were chosen (e.g., prior literature, or trial and error).

• Integrate the supporting information (e.g., S1 Appendix) into the main manuscript where appropriate, particularly in the Methodology section. This will enrich the overall understanding and readability of the paper.

• Details from S1 Appendix describing the network structure should be incorporated into Section 3 (Methodology) for better contextual relevance.

• Discuss how PiCCL performs as the number of views (P) scales to higher values or as the size of the dataset increases. Address any potential bottlenecks or limitations.

• Include the testing details currently in S3 Appendix into Section 4 (Results). This will make the results section more comprehensive and self-contained.

• The manuscript mentions using ResNet-18 as the encoder backbone without explaining why it was chosen. Justify this choice, particularly in the context of its advantages or suitability for the datasets used.

• Currently, only accuracy and training time are reported. Including additional performance metrics will provide a more holistic evaluation of the proposed method.

• The future work section could be elaborated further. Consider including specific directions or potential applications to provide a clearer roadmap for follow-up research.

Reviewer #2: This research paper presents “ PiCCL (Primary Component Contrastive Learning), a self-supervised contrastive learning framework that utilizes a multiplex Siamese network structure consisting of many identical branches rather than 2 to maximize learning efficiency. You benchmarked PiCCL against various state-of-the-art self-supervised algorithms on multiple datasets including CIFAR-10, CIFAR-100, and STL-10. PiCCL achieved top performance in most of our tests, but where PiCCL excels is in the small batch learning scenarios. When using a batch size of 8, PiCCL outperforms the competition by more than 3 percentage points”

Good work keeps up

besides that, I have few minor comments which could further improve the quality of the manuscript

1. Provide quantitative remarks of the impact of the proposed method in the abstract.

2. need to rewrite clearly the contribution, motivation, challenges, your paper design.

3. it is better to summarize the literature review in table.

4. it is better to include a flow chart / pseudocode for your work.

5. The superiority performance of the proposed method could be achieved at what cost?

6.A detailed analysis of the limitations and potential failure scenarios of the proposed model is missing

6. PLOS authors have the option to publish the peer review history of their article (what does this mean?). If published, this will include your full peer review and any attached files.

Reviewer #1: No

Reviewer #2: No

---

## [Author Response · Author response to Decision Letter 1]

21 Jan 2025

Respond to reviewer 1:

Thank you so much for reviewing our work. The writer of this manuscript have very limited experience on writing academic papers and your comments are truely invaluable. Here's the changes we made in correspondance to each of your comments:

1. Regarding the flow of the manuscript. I have tried to make adjustments to improve the overall flow of the text.

2. Regarding providing explanations when technical term shows up. I have provided brief explanations of ``pseudo-labels", ``pretext task", as well as some other technical terms at their first appearance in this manuscript.

3. Regarding the placement of the plots. I have moved the flow diagram (originally Fig-1) down to the method section. However, I did not change the placement of Fig-2 (now Fig-1) because it corresponds to a paragraph providing the intuition and motivation behind this paper. If you think moving that paragraph to a different section will improve the overall structure, please point that out and I will comply.

4. Regarding figure reference. We have merged the 2 panels of Fig 2 into a single figure.

5. Regarding the choice of hyperparameters. The hyperparameters are either inherited from Barlow-Twins or picked through trial and error. However, when reevaluating the manuscript, we felt that Appendix S2 was too bulky and technical to be integrated into the main text. The information it contains can be easily accessed from the source code. And for readers not familiar with the torchvision package, the hyperparameters contain little to no meanings. Therefore we decided to remove Appendix S2.

6. Regarding integrating the appendixes into the main text. All appendixes are either integrated into the main text or removed. The revised manuscript contains no appendix.

7. Regarding placement of S1 Appendix. S1 appendix has been integrated into the section ``Results".

8. Regarding how PiCCL's performance scales with the number of views (P). We added a new paragraph to the ``discussion" section, ``performance of PiCCL" subsection where we answered this question. Here's the summary: in our tests, PiCCL(2) performs worse than PiCCL(4), and PiCCL(16) did not offer an increase in accuracy beyond PiCCL(8). Thus we reported the results for PiCCL(4) and PiCCL(8) because we felt these 2 are the most relevant.

9. Regarding placement of S3 Appendix. S3 appendix has been integrated into the section ``Results".

10. Regarding why we choose ResNet-18 as our encoder. We chose it because SimCLR and Barlow-Twins used ResNet-50 in their original publication. We do not have enough computing resources to use ResNet-50 so we used the simpler version. We explained our choice in the first paragraph of the section ``Results".

11. Regarding other performance metrics. We added the memory occupation in the ``Results" section.

12. Regarding future work. We expanded the last subsection of the section ``Discussion" and explained both our short and long term objectives.

Respond to reviewer 2:

Thank you so much for reviewing our work. The first authors are relatively new to the field, and your encouragement means a lot to us. Here's the changes we made in correspondance to each of your comments:

1. Regarding qualitative remarks in the abstract. I have added some accuracy scores to the abstract.

2. Regarding the contribution, motivation, challenges, and design. I have made minor additions across the text trying to explain every choice we made.

3. Regarding using a table to summarize the literature reviews. After consideration, we are worried that the literature mentioned in the section ``Related Works" would not present well in a table since their major differentiating factors are either in the network design or in the loss function, both are difficult to showcase in a table.

4. Regarding adding a flow chart or pseudo code. The flow diagram has been moved to the section ``Method" and has new text accompanying it, aiming to improve the clarity of the algorithm.

5. Regarding the computational performance. A few sentences are added to the ``Efficiency of PiCCL" subsection of the ``Discussion" section to discuss the computational cost. We also added memory usage as a new metric.

6. Regarding the limitations. A newly added paragraph in the ``Performance of PiCCL" subsection of ``Discussion" describes how the number of network branches affects performance. It points out that P too small will result in sub-optimal performance and P too large will result in too much computational complexity.

---

## [Decision Letter · Decision Letter 1]

16 Mar 2025

PONE-D-24-42971R1PiCCL: a lightweight multiview contrastive learning framework for image classificationPLOS ONE

Dear Dr. Wang,

Thank you for submitting your manuscript to PLOS ONE. After careful consideration, we feel that it has merit but does not fully meet PLOS ONE’s publication criteria as it currently stands. Therefore, we invite you to submit a revised version of the manuscript that addresses the points raised during the review process.

We look forward to receiving your revised manuscript.

Kind regards,

Hung Thanh Bui, Ph.D

Academic Editor

PLOS ONE

**Additional Editor Comments:**

The revision is better.

In my previous comment, they said that “this is an innovative methodology that, to the best of our knowledge, has not yet been published in any peer-reviewed journal. Our research focuses on comparing transformer-based language models trained in Portuguese and specialized for a specific problem domain”. In this language and domain, maybe there is not any research, but in English there are many research focusing on using transformer models, it’s better they should apply their method and compare with another research on another dataset.

What is their improvement on transformer models, they should discuss in detail.

They should show some cases where their model got the best and worst result and analyze in detail.

Reviewers' comments:

Reviewer's Responses to Questions

**Comments to the Author**

1. If the authors have adequately addressed your comments raised in a previous round of review and you feel that this manuscript is now acceptable for publication, you may indicate that here to bypass the “Comments to the Author” section, enter your conflict of interest statement in the “Confidential to Editor” section, and submit your "Accept" recommendation.

Reviewer #3: (No Response)

Reviewer #4: (No Response)

Reviewer #5: All comments have been addressed

2. Is the manuscript technically sound, and do the data support the conclusions?

Reviewer #3: Yes

Reviewer #4: (No Response)

Reviewer #5: Yes

3. Has the statistical analysis been performed appropriately and rigorously? 

Reviewer #3: Yes

Reviewer #4: (No Response)

Reviewer #5: Yes

4. Have the authors made all data underlying the findings in their manuscript fully available?

Reviewer #3: Yes

Reviewer #4: (No Response)

Reviewer #5: Yes

5. Is the manuscript presented in an intelligible fashion and written in standard English?

Reviewer #3: Yes

Reviewer #4: (No Response)

Reviewer #5: Yes

6. Review Comments to the Author

Reviewer #3: (No Response)

Reviewer #4: (No Response)

Reviewer #5: Thank you to the authors for their effort in revising the paper. The improvements made in this version have strengthened the manuscript significantly. The paper can now be considered for acceptance, provided that the following minor comments are addressed.

Comment #1 - Abstract:

The abstract claims that PiCCL achieves top performance in most tests, but it lacks a thorough statistical significance analysis to validate these claims. Providing confidence intervals or statistical significance tests (e.g., p-values) would strengthen the credibility of the reported performance improvements.

Comment #2 - Introduction:

The introduction does not clearly state the specific gaps in existing literature that PiCCL aims to address. While it discusses the advantages of PiCCL, it does not explicitly compare the limitations of prior methods, making it difficult to see the novelty of the proposed approach.

Comment #3 - Related Works:

To strengthen the discussion on the capabilities of deep learning in image classification, I recommend citing relevant works that highlight the effectiveness of deep learning in real-world applications. Specifically, the authors may consider including the following references:

Applying Image Processing and Computer Vision for Damage Detection in Photovoltaic Panels (DOI: 10.58491/2735-4202.3263)

Automated Defect Detection in Solar Cell Images Using Deep Learning Algorithms (DOI: 10.1109/ACCESS.2024.3525183)

A Review on Detection of Solar PV Panels Failures Using Image Processing Techniques (DOI: 10.1109/MEPCON58725.2023.10462371)

These papers demonstrate the application of deep learning and contrastive learning techniques in image-based fault detection and classification, further reinforcing the significance of self-supervised learning in practical scenarios.

Comment #4 - Conclusion:

The conclusion overstates PiCCL’s potential without acknowledging its limitations. While the paper suggests that PiCCL can be used in online learning, no real-world applications or deployment experiments are conducted to support this claim.

7. PLOS authors have the option to publish the peer review history of their article (what does this mean?). If published, this will include your full peer review and any attached files.

Reviewer #3: No

Reviewer #4: No

Reviewer #5: No

---

## [Author Response · Author response to Decision Letter 2]

7 May 2025

Response to Reviewer 3

Thank you for taking your time reviewing our work. Here's our response to each of your comments.

1. Thank you for your valuable comment. In response, we have picked the 2 tests which we think is the most demonstrative (batch size=8 and 256 on STL-10), and retested it 4 more times (5 times in total) and completed a student T test. We also provided confidence interval on the difference between the accuracy between methods. We added a new figure to display the results (Fig.4), and also added a few paragraphs in section 4.1 explaining it.

2. We added a few sentences throughout the manuscript to provide some background for why we created PiCCL. For example: beginning of introduction paragraph 3, last sentence of last paragraph of section 2.1.

3. Deep learning is indeed very useful in real world applications, this is also one of our future objectives. I have cited the provided paper in section 5.4.

4. Thank you for pointing this out, we have reduced the claims in the conclusion section.

Response to Reviewer 5

1. We added a sentence in the beginning of introduction paragraph 3 to express our research objective. Last sentence of last paragraph of section 2.1 also provides some detail.

2. Thank you for pointing this out, we have added a new paragraph in section 2.1 dedicated to 2 newer algorithms which we think provides some more background for our method.

3. We have accepted your proposition and added a table in the related works section.

4. The encoder network, ResNet-18, is a popular network for image classification and thus we did not provide network structure details for it. The modifications we made to the encoder network, as well as the detailed structure of the projection head, are expressed in the first paragraph of results section.

5. We have added a few sentences to fill in some explanation we previously missed. The networks is explained in the first paragraph of section 4, the loss function is explained in section 3.2, and the image augmentation is explained in section 3.1.

6. A table containing the augmentation methods and its parameters is added to section 3.1.

7. The datasets used in this study (STL-10, CIFAR-10, and CIFAR-100) are all well known datasets. Section 4.1, 4.2 provides some info on these datasets including number of samples, type of samples, subset segmentation. We also added a few sentences explaining how each subset are used in our study.

8. Sorry for the confusion, but most unsupervised contrastive learning methods (including PiCCL) wouldn't work without data augmentation because the generation of positive sample pairs rely on data augmentation. For supervised learning, a positive pair can be obtained by selecting 2 samples with the same label. Unsupervised contrastive learning methods obtain positive sample pairs by randomly augmenting a sample 2 times. The network doesn't know what the sample contains, but the 2 augmented views from the same sample should contain the same subject. Without augmentation, the 2 views would be identical.

9. Thank you for pointing this out, we have added more captions to make the figures and tables more informative.

10. We have added some analysis on time complexity in section 5.2., Table 1 also lists the time complexity of related methods.

11. Thank you pointing out the lack of statistical analysis and providing the paper. We have decided to employ a student T test for the 2 most representative tests. We also provided the confidence interval for the difference between accuracies. However we think the provided paper strengthens the claim that machine learning is very successful in real world applications, and decided to cite it in section 5.4.

12. A table is added in the related works section to compare the relevant existing works.

13. we think the provided papers strengthens the claim that machine learning is very successful in real world applications, and have cited it in section 5.4.

---

## [Decision Letter · Decision Letter 2]

14 May 2025

PONE-D-24-42971R2PiCCL: a lightweight multiview contrastive learning framework for image classificationPLOS ONE

Dear Dr. Wang,

Thank you for submitting your manuscript to PLOS ONE. After careful consideration, we feel that it has merit but does not fully meet PLOS ONE’s publication criteria as it currently stands. Therefore, we invite you to submit a revised version of the manuscript that addresses the points raised during the review process.

We look forward to receiving your revised manuscript.

Kind regards,

Hung Thanh Bui, Ph.D

Academic Editor

PLOS ONE

**Additional Editor Comments:**

I overlooked some points in my initial review, which is why I have more comments this time. There are some points the authors should take care of as follows:

- The authors should cite all related works in the Table 1.

- In this research, the authors chose multiview contrastive learning algorithms, they should explain in detail why they did that.

- Also they used Siamese networks, they should give a reason to do that.

- They said that “image augmentation methods is crucial for the quality of representation learning . A good augmentation should retain as much relevant information as possible while altering the rest”, so they should explain in detail why they chose the image augmentation process used by Barlow-Twins.

- The main contribution is a loss function designed for PiCCL, they should explain it in detail.

- They chose L2 normalized embedding vector in their loss function, why did they do that.

- In comparison, they only compared with SimCLR ([12]:2021), Barlow-Twins ([14]:2021) and SimSiam ([13]:2020)), how did they get the result of other works?

- What reason did their proposed model PiCCL (8) get the best result, could they explain in detail.

- Take a look the result in Table 6, PiCCL (8) is not good in both Time per epoch and Memory, what is a reason?

- They should do more experiments, analyze the result in detail and compare with advanced methods in recent year.

Reviewers' comments:

Reviewer's Responses to Questions

**Comments to the Author**

1. If the authors have adequately addressed your comments raised in a previous round of review and you feel that this manuscript is now acceptable for publication, you may indicate that here to bypass the “Comments to the Author” section, enter your conflict of interest statement in the “Confidential to Editor” section, and submit your "Accept" recommendation.

Reviewer #3: All comments have been addressed

Reviewer #4: (No Response)

2. Is the manuscript technically sound, and do the data support the conclusions?

Reviewer #3: Yes

Reviewer #4: (No Response)

3. Has the statistical analysis been performed appropriately and rigorously? 

Reviewer #3: Yes

Reviewer #4: (No Response)

4. Have the authors made all data underlying the findings in their manuscript fully available?

Reviewer #3: Yes

Reviewer #4: (No Response)

5. Is the manuscript presented in an intelligible fashion and written in standard English?

Reviewer #3: Yes

Reviewer #4: (No Response)

6. Review Comments to the Author

**Reviewer #3: **After reviewing the amendments made by the authors that dealt with most of the reviewers' comments, the research appears better than the previous one. Accordingly, there is no objection to accepting the research, for the possibility of publishing it in the journal without additional modifications.

**Reviewer #4:** (No Response)

7. PLOS authors have the option to publish the peer review history of their article (what does this mean?). If published, this will include your full peer review and any attached files.

Reviewer #3: No

Reviewer #4: No

---

## [Author Response · Author response to Decision Letter 3]

27 Jun 2025

No reviewer commented for this revision, therefore no response can be provided.

---

## [Decision Letter · Decision Letter 3]

15 Jul 2025

PiCCL: a lightweight multiview contrastive learning framework for image classification

PONE-D-24-42971R3

Dear Dr. Wang,

We’re pleased to inform you that your manuscript has been judged scientifically suitable for publication and will be formally accepted for publication once it meets all outstanding technical requirements.

Kind regards,

Hung Thanh Bui, Ph.D

Academic Editor

PLOS ONE

Additional Editor Comments (optional):

I accept the revision.

Please check all Tables, Figures, Formulas, language and format of the paper.

Reviewers' comments:

Reviewer's Responses to Questions

**Comments to the Author**

1. If the authors have adequately addressed your comments raised in a previous round of review and you feel that this manuscript is now acceptable for publication, you may indicate that here to bypass the “Comments to the Author” section, enter your conflict of interest statement in the “Confidential to Editor” section, and submit your "Accept" recommendation.

Reviewer #3: (No Response)

Reviewer #4: (No Response)

2. Is the manuscript technically sound, and do the data support the conclusions?

Reviewer #3: (No Response)

Reviewer #4: (No Response)

3. Has the statistical analysis been performed appropriately and rigorously? 

Reviewer #3: (No Response)

Reviewer #4: (No Response)

4. Have the authors made all data underlying the findings in their manuscript fully available?

Reviewer #3: (No Response)

Reviewer #4: (No Response)

5. Is the manuscript presented in an intelligible fashion and written in standard English?

Reviewer #3: (No Response)

Reviewer #4: (No Response)

6. Review Comments to the Author

Reviewer #3: (No Response)

Reviewer #4: (No Response)

7. PLOS authors have the option to publish the peer review history of their article (what does this mean?). If published, this will include your full peer review and any attached files.

Reviewer #3: No

Reviewer #4: No

---

## [Editor Report · Acceptance letter]

PONE-D-24-42971R3

PLOS ONE

Dear Dr. Wang,

I'm pleased to inform you that your manuscript has been deemed suitable for publication in PLOS ONE. Congratulations! Your manuscript is now being handed over to our production team.

Kind regards,

on behalf of

Dr. Hung Thanh Bui

Academic Editor

PLOS ONE